# Cross-species oncogenic signatures of breast cancer in canine mammary tumors

Tae-Min Kim[1], In Seok Yang[2], Byung-Joon Seung[3], Sejoon Lee[4], Dohyun Kim[5], Yoo-Jin Ha [2], Mi-kyoung Seo[2], Ka-Kyung Kim[2], Hyun Seok Kim [6], Jae-Ho Cheong[2], Jung-Hyang Sur[3], Hojung Nam[5] & Sangwoo Kim [2✉]

Genomic and precision medicine research has afforded notable advances in human cancer treatment, yet applicability to other species remains uncertain. Through whole-exome and transcriptome analyses of 191 spontaneous canine mammary tumors (CMTs) that exhibit the archetypal features of human breast cancers, we found a striking resemblance of genomic characteristics including frequent *PIK3CA* mutations (43.1%), aberrations of the PI3K-Akt pathway (61.7%), and key genes involved in cancer initiation and progression. We also identified three gene expression-based CMT subtypes, one of which segregated with basal-like human breast cancer subtypes with activated epithelial-to-mesenchymal transition, low claudin expression, and unfavorable disease prognosis. A relative lack of *ERBB2* amplification and Her2-enrichment subtype in CMT denoted species-specific molecular mechanisms. Taken together, our results elucidate cross-species oncogenic signatures for a better understanding of universal and context-dependent mechanisms in breast cancer development and provide a basis for precision diagnostics and therapeutics for domestic dogs.

[1] Department of Medical Informatics and Cancer Research Institute, College of Medicine, The Catholic University of Korea, Seoul 06591, South Korea. [2] Department of Biomedical Systems Informatics and Brain Korea 21 PLUS Project for Medical Science, Yonsei University College of Medicine, Seoul 03722, South Korea. [3] Department of Veterinary Pathology, Small Animal Tumor Diagnostic Center, College of Veterinary Medicine, Konkuk University, Seoul 05029, South Korea. [4] Department of Pathology and Translational Medicine, Seoul National University Bundang Hospital, Seongnam 13620, South Korea. [5] School of Electrical Engineering and Computer Science, Gwangju Institute of Science and Technology (GIST), Gwangju 61005, South Korea. [6] Severance Biomedical Science Institute, Yonsei University College of Medicine, Seoul 03722, South Korea. ✉email: swkim@yuhs.ac

Cancer arises in dogs of all ages, just as in humans. Unlike commonly used animal models with artificial genetic modifications, canine tumors occur spontaneously with an intact immune system in ordinary living environments[1]. Moreover, in addition to similarities in anatomy and physiology between dogs and humans, canine tumors also exhibit the principal pathologic features of human cancers, including a long-term oncogenic setting, intratumoral heterogeneity, acquired resistance to treatment, and distant metastases[2]. Hence, canine tumors are invaluable representatives for human cancer research[3,4]. Among canine tumors, canine mammary tumors (CMTs) are the most common in female dogs[5] and have been studied for a long time[6]. CMTs share molecular and clinical features with human breast cancers[7], providing a basis for the adoption of classification systems including genetic, morphological, and prognostic elements[8]. Nowadays, calls for a deeper understanding of the molecular characteristics of CMTs are growing in order to uncover cross-species hallmarks of cancer and to provide better opportunities for treating cancers in dogs.

Despite their apparent similarities, CMTs and human breast cancers show molecular and histological discrepancies that have perplexed veterinary and cancer researchers. For example, unlike in human breast cancers, the clinical benefits of *Her2* amplifications and their association with Her2 overexpression are not straightforward in CMTs[9,10], putting into question the incidence and potential clinical utility of *ERBB2* amplification in CMTs. Moreover, the histological features of CMTs differ from those of human breast cancer. For example, benign tumors are more prevalent in CMTs (i.e., half of the observed cases)[11]. And tumors with mesenchymal origins (e.g., fibrosarcomas and carcinosarcomas) and proliferation of myoepithelial cells (e.g., complex adenomas/carcinomas) are often found in CMTs, all of which are extremely rare in human breast cancers[12]. These observations may imply the presence of distinctive mechanisms underlying carcinogenesis and cancer progression in and the need for more-specified therapeutic strategies for CMTs.

Several studies have attempted to advance understanding of the genetic landscape underlying CMTs. Beck et al.[13] documented CMT-specific gene fusions and deletions using low-depth genome sequencing of five cases. Gene expression profiling revealed genetic markers of disease progression and locoregional metastasis[14,15]. More recently, Liu et al.[16] employed whole-exome sequencing (WES) and RNA sequencing (RNA-seq) of 12 CMT cases to identify histology-specific genetic alterations in CMTs: the authors proposed somatic alterations and epigenetic alterations as markers for simple and complex carcinomas, respectively. However, the mutational landscape of CMTs remains somewhat unclear owing to the small cohort sizes and lack of integrative analysis in these studies. We presumed that multi-omics profiling of CMTs in a large cohort, as in research into human cancers, would lead to better understanding of the underlying molecular pathogenesis of CMTs and inter-species relationships with human cancer.

Here, we report our analysis of WES and transcriptome-sequencing (RNA-seq) data for 191 CMT cases, as the first cohort-level multi-omics study in canine cancers. Our study covers most of the latest genomic analyses applied in human cancer research, including the landscape of somatic mutations and involved pathways, mutational features (mutation burden and signatures), clonal selection, subtype specificity, gene expression, molecular subtyping, immune microenvironment, and survival analysis. We show a notable similarity between CMT and human breast cancers in terms of recurrent aberration in oncogenic pathways, which suggests molecular convergence of carcinogenesis, and highlight a number of novel CMT-specific mutations and their effects on tumor characteristics. Inclusion of a substantial number of benign tumors, which are usually not available in human, was relied upon to identify oncogenic characteristics in early cancer development. Finally, our study outlines molecular subtypes of prognostic relevance and suggests a need for the discovery of novel biomarkers of CMTs with which to facilitate early diagnosis for curative surgery and to develop targeted therapies.

## Results

**Research cohort**. All CMT specimens were obtained from 191 female dogs after curative surgery. Clinicopathological information for the cohort is summarized in Supplementary Data 1. The cohort comprised three histology types in 43 benign tumors (17 simple and 15 complex adenomas and 11 benign mixed tumors) and >5 histology types in 148 malignant tumors (78 simple and 44 complex carcinomas, 17 carcinomas in benign mixed tumors, and 9 others, including 4 osteosarcomas and 3 carcinosarcomas). The most frequent histologic type in the cohort was simple carcinoma, 63% (49/78) of which were of tubulopapillary type, resembling the natural incidence of malignant CMT[8]. The average age at diagnosis was 11.8 years. In total, WES data for 183 cases (all with matched blood sequencing data) and RNA-seq data for 157 cases (64 cases with matched normal tissue sequencing data) were obtained and further analyzed to ascertain the genomic and transcriptomic landscape of CMT. Sequencing information for WES and RNA-seq are available in Supplementary Data 2 and 3, respectively.

**Somatic mutation profiles of the CMT genome**. By comparing the tumor and matched normal sequencing data, we identified two types of somatic alterations: single-nucleotide substitution/variations (SNVs) and short insertions/deletions (indels). A total of 10,855 exonic mutations (8569 SNVs and 2286 indels, Supplementary Data 4) were identified for 183 cases with WES by a customized variant calling pipeline that includes strict filtration and canine-specific annotation (see Methods). The mutation landscape for nine recurrently mutated genes in CMT (eight genes harboring non-silent mutations for >5% of cases and *AKT1*) is depicted in Fig. 1a.

Among the somatic mutations, *PIK3CA* mutations were the most frequent (91 missense mutations and five in-frame indels in 43.1% of all 183 cases), which is consistent with activation of the PI3K-Akt pathway in breast cancers[17,18]. The majority of the missense mutations occurred at hotspot positions. The most frequently mutated amino-acid residue was H1047R/L (65.9% of 91 missense mutations); 12 other missense mutations (6.5%) were also observed on known *PIK3CA* hotspots[19], suggesting that *PIK3CA* mutations are likely activating mutations and functional drivers of CMTs (Fig. 1b). Mutations on other genes in the PI3K-Akt pathway were also frequently observed: *PTEN* mutations (nine missense, two nonsense, and thee frameshift indels; 6.5% of cases), *PIK3R1* mutations (two missense, one splicing and two frameshift; 6.0% of cases), and *AKT1* mutations (eight missense, all on the known E17K hotspot; 4.9% of cases)[20] (Fig. 1b). Overall, 55.7% of the CMTs harbored at least one non-silent mutation in four PI3K-Akt pathway genes. Of note, although most of the mutations within the PI3K-Akt pathway were widespread across different histology types, *AKT1* mutations were exclusively observed in complex carcinomas (0/78 vs 8/44 in simple and complex carcinomas, respectively; $P = 0.0002$, Fisher's exact test), suggesting a tissue-specific role of the mutation.

Recurrent mutations in CMT outside the PI3K-Akt pathway showed intertumoral heterogeneity. *KRAS* mutations were found in 19 cases (10.4%), a relatively higher rate than that in human

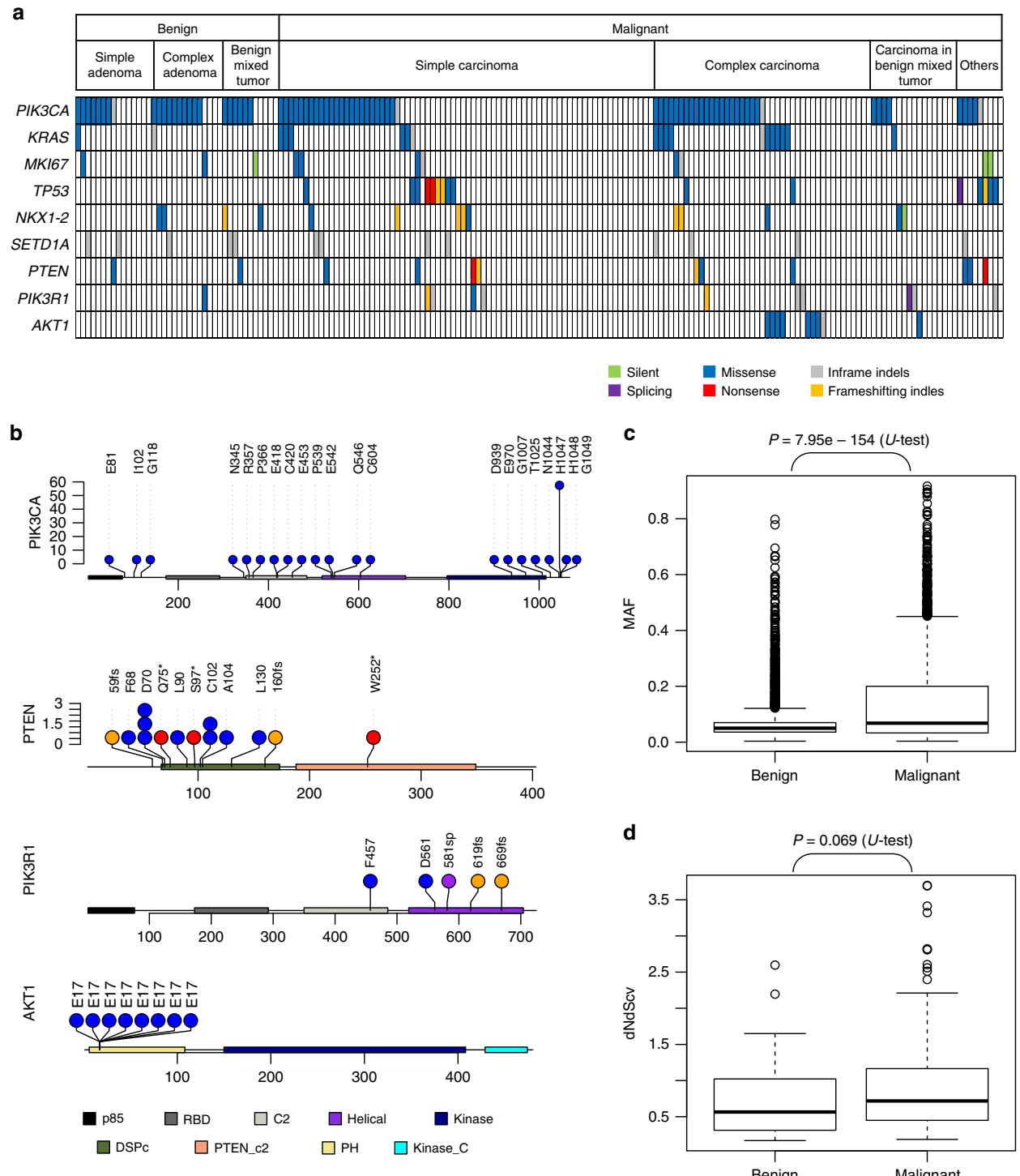

**Fig. 1 Landscape of somatic mutations in CMT. a** The mutational landscape of 183 CMTs (40 benign and 143 malignant CMTs) are shown for nine recurrently mutated genes. For benign and malignant CMTs, major histology types are shown at the top. Six mutation classes with respect to functional changes in the encoded amino acids are shown according to the color legend. **b** Non-silent mutations in four PI3K-Akt pathway genes are shown in lollipop plots. **c** Mutant allele frequencies of 3968 and 497 mutations in benign and malignant CMTs are compared. **d** dNdScv values are compared between 38 benign and 136 malignant CMTs. The significance was estimated by two-sided $U$ tests. For all boxplots in this manuscript, the median and 1st/3rd quartiles of the data were plotted as the center line and the lower/upper boundaries. Whiskers represents the minimum and maximum of the data after removing outliers, which were defined as values smaller than 1st quartile – 1.5× IQR (interquartile range) or larger than 3rd quartile + 1.5× IQR.

breast cancers (5%)[21]. All observed SNVs in *KRAS* were located on three known hotspots: 14 p.G12D/V/A substitutions, p.G13C, and p.E63K, with no silent mutations. This indicates *KRAS* mutation as another major driver of CMT. Truncating mutations

in *NF1* (one frameshift indel and two splicing mutations with two missense mutations) and in *SF3B1* (one splicing mutation) were also observed in CMT, findings that are consistent with truncating mutations in human breast cancers (1.5% of putative

**Table 1 Germline variants of CMT in DNA damage-repair pathways.**

| Function | Genes | P value | Genes with predisposing germline variants |
|---|---|---|---|
| Homology-dependent recombination (HDR) | 88 | 0.029 | *BRCA1, BRCA2, NBN, NSMCE1, POLD1, RECQL4, RMI1, RTEL1, SLX4, SMC5, TOP3B, XRCC3* |
| Fanconi anemia (FA) | 41 | 0.081 | *BRCA1, BRCA2, FAN1, RMI1, SLX4, TOP3B* |
| Direct repair (DR) | 4 | 0.266 | *MGMT* |
| Non-homologous end joining (NHEJ) | 23 | 0.519 | *NBN, RIF1* |
| Mismatch repair (MMR) | 24 | 0.542 | *PMS1, POLD1* |
| Base excision repair (BER) | 47 | 0.689 | *APEX1, NEIL3, POLD1* |
| Translesion synthesis (TLS) | 20 | 0.787 | *POLI* |
| Nucleotide excision repair (NER) | 51 | 0.901 | *ERCC6, POLD1* |
| Nucleotide pools (NP) | 5 | 1.0 | — |

The level of significance (P value) was estimated with Fisher's exact test.

driver mutations and mutations of unknown significance in *NF1* and *SF3B1*, respectively)[21].

We further found that novel recurrent mutations at hotspot sites may account for species-specific mechanisms of carcinogenesis and progression in CMT. Encoding Ki-67 protein, a proliferation marker, *MKI67* was frequently mutated in our cohort (6.0% cases; six missense mutations and two in-frame indels). Although Ki-67 protein levels have been proposed as a proliferation marker for discriminating luminal A and luminal B subtypes, no recurrent *MKI67* mutations have been reported in human breast cancers. The recurrent nature of *MKI67* mutations in CMT, as well as the presence of mutation hotspots (e.g., six non-silent mutations occurred at a single amino-acid residue of p. C1606), is indicative of a potential oncogenic role for *MKI67* mutations in CMTs.

**Germline predisposing variants in CMT**. To identify predisposing genetic events in CMT genomes, we detected 2005 germline variants (1124 SNVs and 881 indels) that are uncommon (novel or minor allele frequency <5%) and potentially damaging (i.e., truncating, likely pathogenic or pathogenic) (Supplementary Data 5). First, we found 10 cases harboring germline predisposing variants in *BRCA1/2* genes, the prevalence of which (5.5%, 10/183) is slightly higher than those of human cancers (2.9–3.0%), measured in a meta-analysis of unselected breast cancer patients[22]. Among six *BRCA1/BRCA2* mutations observed, four (two *BRCA1* and two *BRCA2* mutations) were stop-gain or nonsense mutations indicative of the potential functionality. Of interests, five out of the 10 *BRCA1/2* germline variants harboring cases were observed in complex carcinomas suggestive of tissue type-specific mutations along with *AKT1* somatic mutations (Supplementary Fig. 1). High prevalence of *BRCA1/2* germline has been reported in basal-like and triple-negative breast cancer (TNBC) in human (14–15%)[22], suggesting that the impact of inherited deficiency of *BRCA1/2* is not universal, but context-specific in both species.

Mutation enrichment analysis on nine DNA damage-repair pathways[23] further identified that genes harboring germline variants are significantly enriched in homology-dependent recombination (HDR) pathway (*P* = 0.029; Fisher's exact test, see Table 1). Other than *BRCA1/2*, 10 additional mutations were observed in the genes involved in HDR pathway (*NBN, NSMCE1, POLD1, RECQL4, RMI1, RTEL1, SLX4, SMC5, TOP3B,* and *XRCC3*), which may be also responsible for the CMT pathogenesis as shown in the example of *NBN* germline variants and associates breast cancer risk[24].

**Global patterns of somatic mutations**. Analyzing tumor mutation burden (TMB) (i.e., number of exonic mutations in a given genome), we recorded 10–198 exonic mutations per case of CMT

(median of 30 and mean of 43.5 exonic mutations per case), except for an outlier with 2939 mutations. Compared with CMT, human breast cancers[21] showed significantly higher level of TMB overall, i.e., *n* = 981 cases in the TCGA consortium; 45 median and 92.8 mean exonic mutations per case (*P* = 1.9e-9, *U* test; Supplementary Fig. 2).

The mutation spectra and the functional consequences of mutations (e.g., changes in coding residues) were overall constant among CMTs (Supplementary Fig. 3). Considering the outlier case (CMT-033; 2939 exonic mutations) as hypermutated, the frequency of hypermutation in CMT (0.54%; 1 out of 183) was lower than that in human breast cancer (2.03%, 20 out of 981; cutoff for hypermutation >10 mutations per Mb). Although no mutations were observed in proofreading DNA polymerases (*POLD1* and *POLE*) in CMT-033, we observed one truncating somatic mutation in *MUTYH* and missense mutations in genes encoding DNA damage-repair pathways, such as *LIG1, LIG3, XRCC5, BRCA2,* and *XPC* (Supplementary Data 4), which may contribute to a mutator phenotype for the CMT genome. In addition, 43 germline predisposing variants were observed in CMT-033 but no variants were found in DNA damage-repair pathways (Supplementary Data 5).

Next, we examined the base substitution patterns of the somatic mutations based on the frequencies of 96 trinucleotides. We found that the trinucleotide frequencies were mostly similar across the examined CMT cases (Supplementary Fig. 4a). We first employed de novo mutation signature deconvolution based on non-negative matrix factorization (NMF) and Wellcome Trust Sanger Institute (WTSI) mutation signature framework (https://kr.mathworks.com/matlabcentral/fileexchange/38724-sigprofiler)[25,26]. Both methods revealed mutation signatures similar to Signature #1 (Sanger ver. 2 30 COSMIC mutation signatures #1 to #30, cosine similarity of 0.74–0.86) (Supplementary Fig. 4b, c) suggesting Signature #1 represents a major mutation signature in CMT genomes. We further employed mutation signature assignment analyses to estimate the levels of 30 COSMIC mutation signatures for all CMT genomes. Consistently, we observed that a single mutation signature of Signature #1, which represents age-related mutations, were prevalent in all CMT genome profiles (Supplementary Fig. 4d). These findings indicate that the mutation forces active during the carcinogenesis of CMT are largely uniform across individual cases and highlight spontaneous deamination leading to C-to-T transitions at CpG dinucleotides as the major contributor of somatic mutations therein (Supplementary Fig. 4c).

**Mutations in early cancer development and progression**. Inclusion of multi-type benign tumors in the cohort (43 out of 191, 22.5%) allowed us to directly compare genomic mutation profiles between benign and malignant tumors, which has rarely been undertaken in human cancer studies. As shown in the

mutation profiles (Fig. 1a), *PIK3CA* mutations were frequently found in the benign tumors, suggesting that this key driver mutation is commonly acquired in advance of malignant progression. In contrast, *TP53* mutations were found only in malignant CMTs (0 out of 43 benign vs 16 out of 148 malignant CMTs; $P = 0.025$, Fisher's exact test). This suggests that *TP53* mutations may arise as late evolutionary events in malignant progression, instead of CMT-initiating drivers. Likewise, *KRAS* mutations were over-represented in malignant tumors, particularly in complex carcinoma, being indicative of late events in the transformation to mesenchymal phenotypes[27].

We further examined and compared mutation abundance and composition between the benign and malignant tumors. The TMBs of the benign and malignant tumors were similar ($P = 0.60$, *U* test; Supplementary Fig. 2). Given the tumor purity has been proposed as a confounding factor in evaluating TMB, we further adjusted TMB with respect to the estimated tumor purity[28]. Consistently, no statistically significant difference was observed between the purity-adjusted TMB of benign and malignant CMTs ($P = 0.44$, *U* test). These findings suggest that the abundance of mutations largely remain the same during malignant progression. However, two mutation-based measures associated with tumor evolution (mutant allele frequencies (MAFs) and sample-wise dNdScv scores[29] (dN/dS ratio)) provided clues on potential transforming events during benign-to-malignant progression. We found that the MAFs of malignant CMTs were significantly higher than those of benign CMTs ($P = 7.95e-154$; *U* test) (Fig. 1c), indicating that the malignant progression of CMT may accompany clonal selection events affecting subclonal mutation architecture, such as clonal sweeps. In addition, sample-wise dNdScv scores reflective of degrees of positive selection on individual cases[29] were also substantially higher in malignant CMTs than in benign CMTs (mean dNdScv score of benign and malignant CMTs being 0.73 and 0.91, respectively) although not significantly different ($P = 0.069$; *U* test) (Fig. 1d). This demonstrated that the mutation composition of malignant CMTs shifts towards non-synonymous mutations, making them more likely to endure positive selection, compared with benign CMTs[29]. Gene-wise application of dNdScv scores revealed that the top 14 genes with positive selection (false discovery rate or FDR < 0.3; Supplementary Table 1) included four genes in the PI3K-Akt pathway (*PIK3CA*, *PIK3R1*, *PTEN*, and *AKT1*) and frequently mutated genes, such as *KRAS* and *TP53*. In particular, no silent or synonymous mutations were observed in the four PI3K-Akt pathway genes. Taken together, the molecular mechanisms underlying the progression of CMT can be modeled as (i) the emergence of early oncogenic mutations (e.g., *PIK3CA* mutation) in benign tumors, (ii) subsequent acquisition of additional drivers accompanying malignant transformation (e.g., *TP53* and *KRAS* mutations), and (iii) clonal domination of malignant subclones with genomic footprints (e.g., MAF and dN/dS ratios).

**Somatic copy number aberrations in CMT.** Somatic copy number alterations (SCNAs) were profiled according to the read depth ratios of tumor and matched normal sequencing data. Genome-wide profiles of chromosomal copy number gains and losses in CMT are depicted across histologic types in Fig. 2a (shown in the order of the cases in Fig. 1a). Among the histologic types, SCNA-frequent cases were enriched in simple carcinoma, which is consistent with a previous report[16]. Genomic fractions with copy number imbalances were significantly higher in malignant CMTs than in benign CMTs ($P = 0.0011$; *U* test, Fig. 2b), suggesting that the genomic instability leading to SCNA is a late evolutionary event occurring after malignant progression.

We also observed that genomes harboring *TP53* mutations more commonly had SCNAs than genomes that did not ($P = 8.8e-06$; *U* test, Fig. 2b), supporting the notion that *TP53* mutations lead to the accumulation of genomic instability and aneuploidy in cancer genomes[16].

In regards to specific genes, recurrent SCNAs were observed on *AKT1* (amplified in 2.7%) and *PTEN* (deleted in 10.9%). Approximately two-thirds of the cases (113 CMTs, 61.7% of the cohort) showed genomic aberrations in PI3K-Akt pathway genes, harboring either somatic mutations or SCNAs in *PIK3CA*, *PTEN*, *PIK3R1*, and *AKT1*. We further performed GISTIC analysis to identify 18 recurrent amplification and 49 recurrent deletion peaks across the CMT genomes (Supplementary Table 2). The identified GISTIC peak regions mirrored known cancer-related genes (i.e., Cancer Gene Census[30]) (Fig. 2c). Among the recurrently amplified and deleted loci, we noted that canonical oncogenes, such as *EGFR* and *HRAS*, were recurrently amplified in CMTs; however, amplification of *ERBB2* and *MYC*, which is common in human breast cancers, was not reflected in the GISTIC peaks of CMTs. Thus, we further examined the genomic amplification patterns of *EGFR* in comparison to those of *ERBB2* and *MYC* (Supplementary Fig. 5). In the case of *EGFR*, minimal amplification patterns around the genetic locus were observed and exhibited typical alteration patterns of functional oncogenes (Supplementary Fig. 5A). However, *ERBB2* amplification was often observed as arm-level events or separate from *pter*-located GISTIC peaks, including *ASPSCR1* and *RNF213* (Supplementary Fig. 5B), which is in stark contrast to the minimal amplification of *ERBB2* observed in human breast cancers (Supplementary Fig. 5B, inlet). Together with the low concordance between the copy numbers and gene expression levels of *ERBB2* in CMT ($r = 0.09$, measured in this cohort), unlike in human breast cancers ($r = 0.857$), we can assume a reduced role and limited clinical value for *ERBB2* in CMT, as was previously proposed[9,10]. Similarly, *MYC* amplification was frequently present at the chromosomal level, which resulted in an inability to detect *MYC* by GISTIC peaks (Supplementary Fig. 5C). Also, low correlation between copy numbers and gene expression levels ($r = 0.093$) was observed for *MYC* in CMT, in contrast to human breast cancers ($r = 0.247$). Regarding other genes, recurrent *TERT* amplification was noted; the prognostic implications thereof have been described for human breast cancers[31]. We also observed recurrent loss of genes with roles in genomic instability, including *ATM* and *TP53*, which are well-recognized cancer-associated genes in human cancers.

**Transcriptome analysis of CMT.** RNA-seq data were available for 157 tumors and 64 matched normal CMTs. We first applied NMF for the CMT transcriptome data set to delineate key transcriptional features or metagene signatures at the bulk level. Cophenetic scores showed that at least five metagene signatures were present in the transcriptome data set (Supplementary Fig. 6). Among the five metagene signatures (annotated NMF1–NMF5), three were relatively specific to CMTs (NMF1, NMF2, and NMF3); the other two signatures (NMF4 and NMF5) were specific to adjacent normal breast tissues (black and gray, respectively; Fig. 3a). We further performed gene set enrichment analysis to identify molecular functions associated with the five metagene signatures (Supplementary Table 3). According to the enriched molecular terms (based on the Hallmark gene set of MSigDB), we could annotate three tumor-specific metagene signatures as "mitosis" (NMF1), "DNA repair" (NMF2), and "epithelial-to-mesenchymal transition" (EMT) (NMF3). Likewise, the other two normal tissue-specific signatures were annotated as "estrogen-late" (NMF4) and "estrogen-early" (NMF5). The

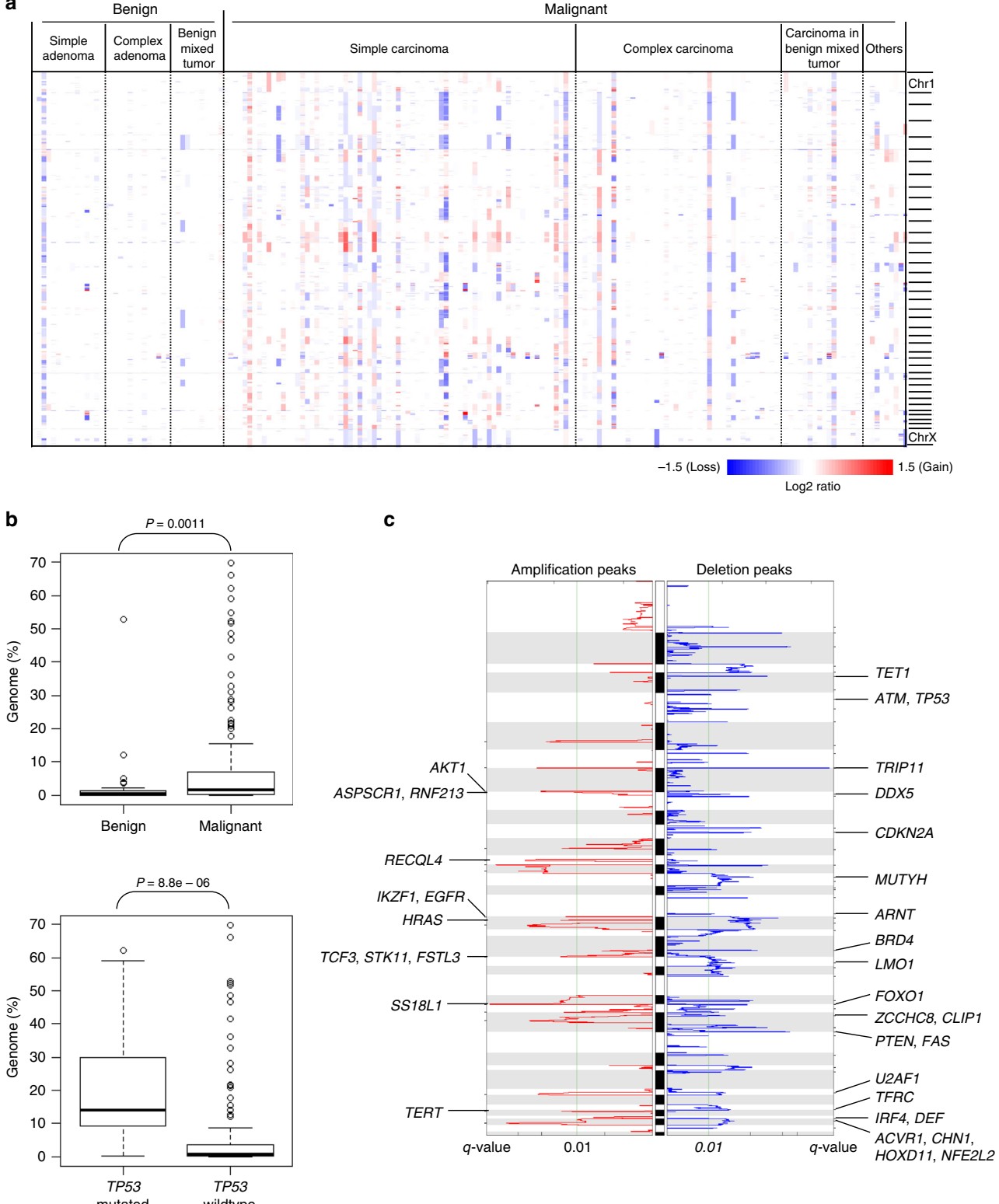

**Fig. 2 Somatic copy number alterations (SCNAs) in CMT genome profiles. a** A genome-wide heatmap of SCNAs is depicted, with red and blue representing chromosomal gains and losses, respectively. The cases are listed in order as in Fig. 1a. **b** Genome fractions with SCNAs were compared between 40 benign and 143 malignant CMTs, as well as between CMTs with or without TP53 mutations ($n = 16$ and 167, respectively). **c** Amplification and deletion peaks identified by GISTIC algorithms are shown across canine genomes with cancer-related genes belonging to the peaks shown. The significance was estimated by two-sided $U$ tests.

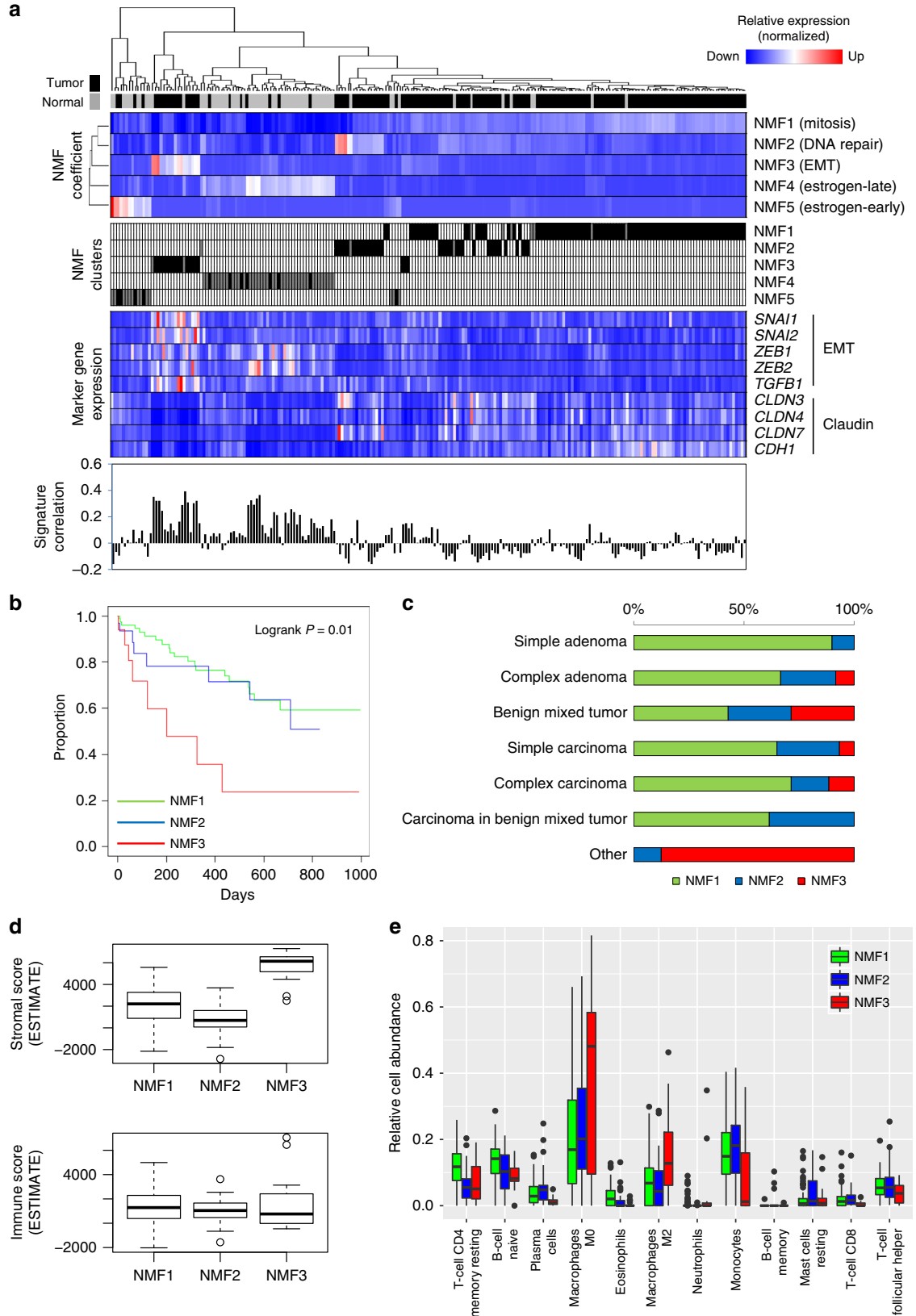

enrichment plots and top-enriched genes (i.e., leading edge genes) for the selected functions of the five metagene signatures are shown in Supplementary Fig. 6. Marker gene expression analysis revealed that NMF3 CMTs showed upregulation of the EMT markers *SNAI1/2*, *ZEB1/2*, and *TGFB1*. Of note, NMF3 CMTs further showed down-regulation of claudin-encoding genes (*CLDN3*, *CLDN4*, and *CLDN7*), as well as E-cadherin (*CDH1*). Along with a high degree of correlation between gene expression in NMF3 CMTs and that in tumor-initiating cells ("Signature correlation" in Fig. 3a), these findings suggest that NMF3 CMTs recapitulate the molecular features of claudin-low breast cancer subtypes[32].

**Fig. 3 Molecular taxonomy and tumor microenvironments of CMT based on transcriptomic analyses. a** Five NMF metagene signatures are identified (NMF1–NMF5) with functional annotations. A heatmap shows that three NMF metagene signatures (NMF1–NMF3) are upregulated to varying degrees in CMT tumors (black), whereas NMF4 and NMF5 signatures are upregulated in normal tissue transcriptomes (gray). NMF clusters for individual CMTs are assigned according to the level of five NMF metagene signatures. The expression of selected EMT markers and claudin genes is also shown in a heatmap. The degrees of correlation between the expression signatures of tumor-initiating cells and CMT transcriptomes are depicted in signature correlation. **b** CMT tumors are classified into three subtypes (NMF1, NMF2, and NMF3), and their respective Kaplan–Meier survival curves are shown. The significance of survival differences across NMF clusters was estimated by log-rank test (two-sided). **c** Three NMF CMT subtypes were compared with histology types. **d** ESTIMATE-derived stromal and immune scores are plotted against the three NMF CMT clusters (92 NMF1, 35 NMF2, and 18 NMF3 CMTs, respectively). **e** Relative abundance of 12 immune cell subtypes estimated by CIBERSORT algorithms are shown for 92 NMF1, 35 NMF2, and 18 NMF3 CMTs.

We further assessed whether the metagene signatures held any prognostic significance. Re-classification of the 157 CMT tumors with the five metagene signatures assigned 145 tumors to one of the three tumor signatures (97, 35, and 18 to NMF1, NMF2, and NMF3, respectively), which were subjected to survival analysis. Log-rank tests revealed significant differences in overall survival across the three CMT clusters ($P = 0.010$, log-rank test; Kaplan–Meyer survival curves in Fig. 3b), with the least favorable prognosis for the NMF3 CMTs with activated EMT. Although evidence of EMT has been reported in a few canine tumors[33,34], our analysis establishes the presence of an EMT subtype and its association with poor prognosis in CMT, as in multiple human cancers[35]. Interestingly, the NMF3 cluster was enriched with rare CMT subtypes, including carcinosarcomas and osteosarcomas (Fig. 3c). As claudin-low subtypes are known to be enriched with tumor-initiating cells or stem cells that can differentiate into either myoepithelial or luminal progenitors in the hierarchy of mammary epithelial development[36], the molecular features of NMF3 CMTs resembling those of claudin-low subtypes may be responsible for the presence of rare CMT histology subtypes with mixed epithelial and mesenchymal components, such as carcinosarcomas. To further evaluate the clinical relevance of NMF3 CMTs in an extended cohort of human breast cancers, we obtained four expression profiles in public database (GSE17907, GSE20711, GSE25066, and GSE31519). Patients with high NMF3 metagene scores showed unfavorable clinical outcomes, i.e., significantly different survival was observed for two of four cohorts (Supplementary Fig. 8). In addition, high EMT scores were observed for those with high NMF3 metagene scores.

Next, we examined relationships among the metagene signatures with intrinsic molecular classification (e.g., luminal A, luminal B, Her2-enriched, and basal-like subtypes) for human breast cancer[37]. To apply the intrinsic human breast cancer subtypes to CMT, we focused on the expression of genes belonging to PAM50 in the expression profiles of human breast cancers and CMT. After merging the expression profiles, hierarchical clustering thereof largely segregated CMT tumors into two classes, one of which segregated with luminal A (non-basal CMTs) and the other with basal-like subtypes of human breast cancers (basal CMT) (Supplementary Fig. 7). We noted that *ERBB2* overexpression was exclusive to Her2-enriched human breast cancers; it was not observed in other human breast cancer subtypes and CMTs. Basal CMT showed significant unfavorable clinical outcome in terms of survival, compared to non-basal CMTs ($P = 0.004$, log-rank test). In addition, basal CMTs were enriched with NMF3 CMTs and rare histological subtypes of CMT. Altogether, application of human breast cancer molecular subtyping to CMTs revealed that a subset of CMTs sharing the clinical behavior and histologic presentation of NMF3 CMTs is transcriptionally similar with basal-like human breast cancer. However, the results did not support the presence of Her2-enriched human breast cancer subtypes in our CMT cohort.

The presence of EMT-related NMF signatures and their associations with CMT suggested that the tumor microenvironment affects CMT pathogenesis. Accordingly, we applied the ESTIMATE algorithm[38] to estimate the relative fraction of stromal and immune cells in the CMT microenvironment. The analysis revealed significantly different ESTIMATE stromal scores among the three NMF classes ($P = 5.18e{-}18$, analysis of variance; ANOVA) (Fig. 3d). Higher stromal scores for NMF3 CMTs than for NMF1-NMF2 CMTs suggested higher degrees of stromal cell infiltration in NMF3 tumors. Accordingly, we deemed that the EMT-representing transcripts in NMF3 tumors may likely be derived from tumor-infiltrating stromal cells instead of tumor cells, consistent with a recent observation in mouse xenograft models[39]. Although no significant differences in ESTIMATE immune scores were observed among the three NMF classes ($P = 0.388$, ANOVA), it is possible that activated EMT may still impact the immune contexture or the immune cell composition in CMT, as only total immune cell counts are reflected in ESTIMATE immune scores. Thus, we further applied the CIBERSORT algorithm to estimate and compare the relative abundance of 22 immune cells among the NMF classes[40]. Twelve immune cell types showed significant differences ($P < 0.05$, ANOVA) among three NMF tumor types, as shown in Fig. 3e. We noted that NMF3 CMTs were relatively depleted of tumor-infiltrating immune cells, such as naive B cells, CD8 T cells, nature killer cells, and monocytes, compared with NMF1 and NMF2 CMTs. The enrichment of M0 and M2 macrophages was observed for NMF3 CMTs, suggesting that macrophages are the major immune component in the microenvironments thereof. We presume that NMF3 CMTs favor the polarization of M0 macrophages to M2 macrophages, which are important in inflammation and tissue repair, instead of M1 proinflammatory macrophages.

**Cross-species genomic alterations of canine and human breast cancers.** For cross-species comparison of pathway-level mutations and SCNAs between CMT and human breast cancer, we examined 13 genes belonging to two signaling pathways (the PI3K-Akt and p53 pathways), particularly in regards to the frequencies of activation or inactivation thereof in benign and malignant CMTs and in human breast cancer (Fig. 4a). Four genes in the PI3K-Akt pathway, *PIK3CA* (55%, 38%, and 39% activated in benign CMT, malignant CMT and human breast cancer, respectively), *PTEN* (4%, 20%, and 13%), *PIK3R1* (2%, 10%, and 8%), and *AKT1* (0%, 9%, and 4%), showed comparable alteration frequencies between benign/malignant CMTs and human breast cancers, indicating that mutations in the PI3K-Akt signaling pathway are conserved across species in breast cancer pathogenesis. In addition, the higher mutation frequencies of *PIK3CA* in benign CMTs, compared with malignant CMTs and human breast cancers, highlight the early oncogenic roles of *PIK3CA* mutations. Other genes showed relatively lower alteration frequencies in CMTs than in human breast cancers, including *TP53* (inactivated in 0%, 15%, and 48% of benign CMTs, malignant CMTs, and human breast cancers, respectively), as well as *EGFR*, *ERBB2*, *ATM*, and *CHEK2*. In addition,

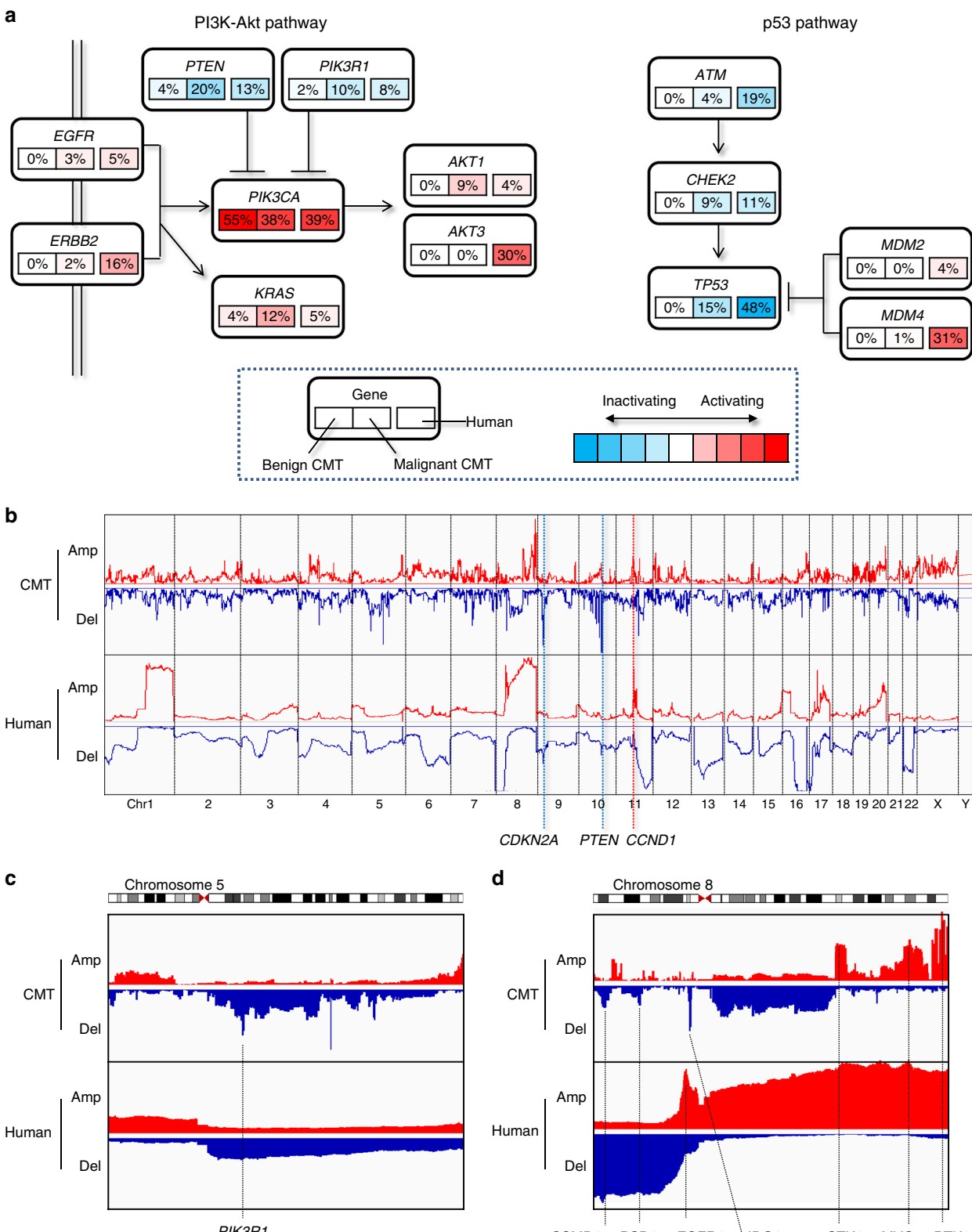

**Fig. 4 Cross-species comparison of mutations and SCNAs. a** For two major signaling pathways (PI3K-Atk and p53 pathways), the respective levels of activation and inactivation (red and blue, respectively) for 13 genes are shown. For each gene, activity levels are shown for benign CMTs, malignant CMTs, and human breast cancers, respectively. **b** CMT SCNAs are aligned onto the human reference genome (hg19) and compared with SCNA profiles of human breast cancers in the linear space of hg19 (chr1 to chr22 with sex chromosomes). The degrees of amplification and deletion are presented as the sum of log-ratios in hg19-aligned CMT and human breast cancer cohorts. **c** Chromosome 5 is shown, where human breast cancer shows arm-level 5p gains and 5q losses. The hg19-aligned CMT genomes show a deletion peak harboring PIK3R1. **d** Chromosome 8 shows peaks in six genes of hg19-aligned CMT SCNAs (CSMD, PSD3, IDO1, STK3, MYC, and PTK2), along with FGFR1 in an amplification peak of human breast cancer SCNAs.

*AKT3*, *MDM2*, and *MDM4* were rarely altered in CMTs. The relative lack of alterations in genes other than PI3K-Atk pathway genes may indicate that a limited repertoire of mutations is sufficient to give rise to CMT in a relatively short time, compared with human breast cancer development. Nevertheless, additional studies are required to investigate whether mutations less frequent in CMT genomes, such as *ERBB2* amplifications, have similar biological or clinical implications as those in human breast cancers or whether they merely represent redundant alterations arising in the background of neutral mutations. The non-silent mutation frequencies of CMT genomes are compared with those of human breast cancers in Supplementary Data 6.

Finally, we transformed the CMT SCNA profiles by synteny realignment (canFam3.1) onto the human reference genome (hg19) using the blastz alignment algorithm[41] (see Methods). SCNA profiles of hg19-aligned CMT genomes and human breast cancers are shown in Fig. 4b. We noted that cross-species correlations between segment-level amplifications and deletions were not strong ($r = 0.225$ and $0.037$ for amplifications and deletions, respectively). However, major peaks in *CCND1* (11q gain), *CDKN2A* (9p loss), and *PTEN* (10q loss) were concordant between dogs and humans. Owing to the different chromosomal constitutions of the two species, a single large (e.g., arm-level) SCNA event in human breast cancers may be represented by multiple SCNA events in CMT, helping to pinpoint regions of functional relevance. In chromosome 5, we observed a narrow deletion peak involving *PIK3R1* in the hg19-aligned CMT SCNAs, which has been shown to reflect arm-level losses of 5q in human breast cancer SCNAs, suggesting functional relevance in terms of PI3K-Akt signaling (Fig. 4c). Likewise, in chromosome 8, arm-level gains of 8q and losses of 8p in human breast cancer SCNAs were further segmented into three amplification peaks involving *STK3*, *MYC*, and *PTK2* and three deletion peaks involving *CSMD1*, *PSD3*, and *IDO1* in hg19-aligned CMT SCNAs, respectively (Fig. 4d). These cross-species comparative oncogenomic results exemplify how genomic analysis of CMT can lead to the better understanding of human breast cancers.

## Discussion
In this study, we conducted a comprehensive genomic analysis of CMT at the cohort level. Exome- and transcriptome-sequencing based molecular characterization revealed somatic mutations and SCNAs in CMTs at an unprecedented scale. Compared with the molecular characteristics of human breast cancers[21], notable similarity in terms of core oncogenic signatures including key genes of the PI3K-Akt and p53 pathways were identified. We were also able to uncover species-specific molecular characteristics, such as uncertain role for *ERBB2* amplification in CMTs, and mutations prevalent in benign tumors that may reflect the early genetic basis underlining the initiation of CMT pathogenesis. Finally, gene expression-based molecular taxonomy revealed the presence of an EMT-associated subtype in CMTs with an unfavorable prognosis. Overall, our study highlights the molecular convergence of key oncogenic pathways and supports the potential use of therapeutics for human breast cancer in dogs with CMTs[42].

Domestic dogs have a much shorter life expectancy than humans (10–13 vs 79 years), and tumorigenesis in dogs is accomplished within a shorter period (~10 years). The relatively shorter period for cancer onset may be responsible for the unique features of CMT. For example, the relative paucity of aneuploidy and SCNA drivers, such as *ERBB2* amplifications in CMT, compared with human breast cancers, may be attributed to the shorter development time of the disease, as aneuploidy has often been considered as a late-stage marker[43]. Less-abundant mutation

burdens of CMT compared with human breast cancers suggests that the mutational composition for breast cancer development may be different across species and those of CMT are relatively simpler compared with human breast cancers. It will require further investigation to see whether the additional mutations acquired by human breast cancer genomes merely represent the driver-accompanying neutral alterations or confer additional benefits. In addition, we found that CMT genomes showed relatively uniform TMB levels and sequence compositions (e.g., mutation signatures). This indicated that the mutational settings giving rise to CMT may be achieved at similar ages.

Of interest, TMB levels were comparable between benign and malignant CMTs in this study. This has also been demonstrated for other tumor types, including colorectal cancers[44], raising two possibilities. The first possibility is that an optimal or tolerable TMB level is fixed for a given cancer cell such that malignant progression allows for only an essential, but limited, number of additional mutations to be acquired. This assumption may be supported by our finding of cross-species similarities in CMT and human breast cancers in terms of PI3K-Akt pathway aberrations. The second possibility is that genetic programs favoring benign and malignant disease are determined early instead of following traditional stepwise acquisition of mutations during disease progression. Nevertheless, since the presence of mutations enriched in malignant disease, such as *TP53* and *KRAS* mutations, supports that the malignant progression of CMT may follow traditional stepwise evolution, determining whether the malignant progression of CMT proceeds in accordance with traditional stepwise or parallel evolution will require further investigation with an extended longitudinal setting (e.g., comparison of multiple samples from the same individual). In addition, we found that measures representative of clone- and gene-level selective forces, such as MAF and dNdScv, indicated that the malignant progression of CMT involves selection events that alter the frequencies and compositions of somatic mutations. Thus, whether such selection events are responsible for or are independent of malignant progression need to be determined.

Our research also raised questions as to whether the intrinsic mechanisms for suppressing cancer development (e.g., DNA repair, cell cycle arrest, or immunosurveillance) are intact or not in CMT. The predominance of a single type of mutation signature (Signature #1) in the CMT genomes suggested that the majority of somatic mutations may be those accumulated during the lifetime of a host, with a limited impact of other mutagenic sources in human breast cancers[45] (e.g., APOBEC overactivity and BRCA deficiency). However, no correlation was observed ($r = 0.04$) between host age and the levels of Signature #1 and it is possible that somatic mutations of CMT genomes have arisen in a limited time period such as a mutational burst, or at least have accumulated in a non-gradual manner probably associated with a loss of DNA repair mechanisms. The identification of susceptibility factors for somatic mutations may also lead to developing means of reinforcing tumor-suppressing systems in both dogs and humans.

As the fierce battle with cancer is now expanding to companion animals, treatment of canine cancer itself is becoming an important issue. Although anticancer agents originally developed to treat human cancers may be applied to dogs, little evidence has been given in terms of their therapeutic efficacy, especially in relation to cost effectiveness. Recently, a study of drug sensitivity showed trametinib (a MEK1/2 inhibitor) to be effective in canine cancer cell lines[46], and we expect more lines of evidence will accumulate on trans-species use of more drugs of these kinds. In spite of a concern regarding the discordance between animal and human in drug efficacy and toxicities[47], treatment of canine cancer may benefit from the development of novel human cancer

drugs that target shared oncogenic mutations (e.g., alpelisib for metastatic breast cancer with *PIK3CA* mutations[48]). We envision that genomic studies of different cancer types, further stratification, and companion diagnostics will lead to more efficient treatment of canine cancers, just as they have done for human cancers over last 10 years.

## Methods

Detailed information on the study design, sample collection, data generation, and quality control strategies has been described in a separate data descriptor paper[49]. Here, we have provided a brief overview of the data that are essential to understanding the presented study: some parts of this section may contain overlapping descriptions with the data descriptor paper, especially for the conventional protocols.

**Cohort design and sample collection**. The cohort was designed as a tumor and matched normal control cohort to facilitate the investigation and comparison of genomic and transcriptomic features. In available cases, blood (buffy coat) and adjacent normal mammary tissues were used as controls for tumor DNA and RNA, respectively. In total, 191 dogs with mammary tumors were recruited via private veterinary clinics in Korea, with informed consent from their owners. Tumor tissues, adjacent normal tissues, and blood were collected from the dogs following the guidelines of the Institutional Animal Care and Use Committee of Konkuk University (KU16106 and KU17162) upon availability. Fresh tissue samples were immediately transferred to RNAlater (Thermo Fisher Scientific, Vilnius, Lithuania), refrigerated overnight at 4 °C, and stored at −80 °C. Genomic DNA was extracted from tumor tissue and buffy coats using QIAamp DNA mini kits (Qiagen, Germany). Total RNA was extracted from tumor and adjacent normal tissues using RNeasy mini kits (Qiagen).

**Histopathology**. For histological examination, sections (4-µm thick) from formalin-fixed paraffin-embedded blocks were stained with hematoxylin and eosin and were diagnosed by two researchers (B.J.S. and J.H.S.). Histological subtyping was based on the World Health Organization classification[50]. The diagnosis of malignancy, which included ambiguous subtypes (e.g., simple adenoma vs simple carcinoma (grade 1), complex adenoma vs complex carcinoma (grade 1), benign mixed tumor vs carcinoma in benign mixed tumor (grade 1)) was determined in accordance with that described by Rasotto et al.[51]. Histological grade was assessed according to the Peña system[52], exclusively on the neoplastic epithelial component. In cases of mammary osteosarcoma and mammary fibrosarcoma, histological grade was assessed according to the grading system for canine osteosarcoma[53] and the grading system for cutaneous and subcutaneous soft tissue sarcoma in dogs[54], respectively. Lymphatic invasion, defined as the presence of tumor cells in peritumoral lymphatic vessels (all cases) and/or regional lymph nodes (only available cases), was also determined.

**WES and RNA-seq**. Among 191 samples, 183 cases with tumor DNA and matched normal DNA were subjected to WES. Two hundred nanograms of fragmented DNA was prepared to construct libraries with the SureSelect Canine All Exon Kit (Agilent, Inc., USA) using the manufacturer's protocol. In brief, qualified genomic DNA samples were randomly fragmented by Covaris, followed by adapter ligation, purification, hybridization, and PCR. Captured libraries were then examined on an Agilent 2100 Bioanalyzer to evaluate quality and were loaded on an Illumina HiSeq sequencer, according to the manufacturers' recommendations. In addition, 157 tumor tissues and 64 matched, normal, adjacent tissues with RNA available were also subjected to RNA-seq. Before library construction, RNA 6000 Nano kits (Agilent Technologies, CA) were used to assess RNA quality. For cDNA library construction, 1 µg of RNA was obtained and purified with oligo-dT magnetic beads. Fragmentation was performed with purified mRNA, and double-stranded cDNAs were synthesized. The cDNAs were primed with poly-A, and sequencing adapters were connected using TruSeq RNA sample prep kits (Illumina, CA). Fragments were filtered to a specific length using BluePippin 2% agarose gel cassettes (Sage Science, MA), and PCR amplification was conducted. Fragment lengths and quality were electrophoretically verified with Agilent High Sensitivity DNA kits (Agilent Technologies, CA). Libraries were observed with a window spanning an average of 392 bp, standard deviation of 66 bp. WES and RNA-seq were performed using Illumina HiSeq 2500 (Illumina, CA) with the protocol out-sourced to Theragenetex Inc.

**Processing of sequencing data**. WES reads were aligned to the CanFam3.1 (*Canis lupus familiaris*) reference genome with BWA-MEM2[55]. Duplicate fragments were marked and eliminated with Picard (version 2.2) (http://picard.sourceforge.net). After assessing mapping quality and filtering out low-quality mapped reads, paired read information was evaluated to ensure that all mate-pair information was in sync between each read. Then, processes of removing PCR duplicates, indel realignment, fixing mate information, base quality score recalibration, and variant quality score recalibration on putative SNVs and indels were performed using

GATK4.0 following GATK Best Practices recommendations[56] with CanFam3.1 (Ensembl Release 91) as a reference. The whole pipeline was implemented in-house[49]. RNA-Seq data of the 157 tumor samples and 64 normal, adjacent samples were mapped to the canine reference genome CanFam3.1 using splice-aware aligner of TopHat[57] (v.2.0.9), with Ensembl gene annotation and fr-firststrand library type. FPKM (fragments per kilobase of transcript per million) values were calculated by Cufflinks[58] (v2.1.1) using aligned bam files.

**Germline variants calling and annotation**. GATK-HC[59] (v4.1.5.0) was used to call germline variants in paired bulk data and filtered by VariantFiltration of GATK4 with the criteria recommended for germline variants; excluding candidates with QD < 2.0, FS > 60.6, MQ < 40.0, ReadPosRankSum < −8.0, MQRanksum <2.5 or MQRankSum >2.5. Variants with ≥10 depth in both or only in normal sample were used for further analysis. Variants were liftovered from CanFam3.1 to hg38 with Crossmap[60] (v0.2.9) using UCSC chain file (camFam3ToHg38.over.chain.gz). Variants successfully liftovered to hg38 were annotated using SnpEff[61] (v4.3t) and ClinVar[62] (build 2020-03-10). We only kept variants that are tagged as truncated (stop gained, splice variant, frameshift), pathogenic, or likely pathogenic. Only novel or uncommon candidates with MAF < 0.05 in Dog Genome SNP Database (DogSD) (release 2017-06-10)[63] from iDOG and within the cohort were included in the final germline variant list.

**Somatic variant calling, filtration, and annotation**. We detected single-nucleotide variations (SNVs) and small insertions/deletions (indels) using Mutect2[64] from GATK4 v4.0.10.1. The VCF file produced by the pipeline utilizes reference bases on the positive strand of CanFam3.1 in the *REF* field, and variants are shown in the *ALT* field. We filtered out falsely detected variants using Filter-MutectCalls from GATK4 and selected PASS variants from the VCF files. Variant Effect Predictor was used to annotate identified variants[65]. To estimate dNdScv scores, we used R packages (https://github.com/im3sanger/dndscv)[29].

We examined the possibility of the somatic mutations being falsely detected by alignment errors. In the absence of utility for alignment error assessment in canine genome, we built our in-house workflow to strictly rule out variant sites with dubious alignment patterns using the aligner-generated alignment scores at optimal and suboptimal mappings. For each SNV position, passing sequence reads with an alternative allele were collected to calculate the alignment score at optimal site (AS), and the alignment score at suboptimal (or secondary) site (XS), which are averaged over all samples to derive the mean alignment scores at optimal site (meanAS) and suboptimal site (meanXS). We judged that the candidate SNVs are likely from alignment errors if the meanXS is greater than or equal to the 80% of the meanAS (meanXS ≥ 0.8×meanAS).

**Tumor purity estimation and TMB adjustment**. Tumor purity of CMT genomes were visually estimated by histological examination of H&E stained slides by a pathologist (B-J Seung). Only samples with >70% of the judged proportion of neoplastic cells were used for analysis. In the case of human breast cancers of TCGA consortium, we obtained consensus purity estimates from a literature[66]. The TMB of individual genomes were multiplied with the correction factor corresponding to the purity of the given case, to derive the purity-corrected TMB[28].

**Copy number variant calling**. We calculated the depth of coverage using GATK and then followed the typical XHMM workflow. SAMTOOLS[67] v1.9 and VARS-CAN[68] v.2.4.3 were used to identify SCNAs following the recommended workflow. First, we ran the *mpileup* function in SAMTOOLS to estimate the bin-level sequencing read depth both for tumor and normal BAM files. The ratios of tumor/normal sequencing read depth were then calculated, and the normalized read depth ratios were further GC-corrected. We applied the circular binary segmentation (CBS)[69] algorithm for segmentation and used the IGV browser for visualization of SCNAs[70]. We also used GISTIC2 to identify recurrent chromosomal gains and losses of CMT genomes[71]. For cross-species comparison of SCNAs between CMTs and human breast cancer, we used the cross-species alignment information of canFam3 and hg19 assemblies generated by the blastz algorithm (http://hgdownload.cse.ucsc.edu/goldenpath/hg19/vsCanFam3/). The segment-level log2 ratios of CMT genomes (canFam3.1) were aligned onto hg19 using the blastz chained alignment. The hg19-aligned SCNA profiles were further smoothed and segmented using the CBS algorithm.

**Transcriptome-based CMT subtypes**. We performed NMF deconvolution to identify latent features in CMT expression profiles by decomposing the log-transformed CMT expression matrix into a basis matrix (hereafter, NMF metagene signatures) and metagene expression profiles[72]. For deconvolution, CMT expression profiles along with those of normal, adjacent mammary tissues were subjected to NMF. To determine the optimal number of NMF metagene signatures, we measured the cophenetic score for 2–10 NMF metagene signatures as a stability measure. Five NMF metagene signatures were derived including three signatures representing CMT tumors. We further performed pre-ranked version of gene set enrichment analysis with functional gene sets (MSigDB, Hallmark category) for functional annotation of metagene signatures[73]. For prognostic evaluation of NMF clusters, Kaplan–Meier survival curves were drawn and log-rank tests were applied

for three tumor-specific NMF clusters. We also performed molecular classification of CMT transcriptomes using PAM50 genes. The expression levels of PAM50 genes for TCGA human breast cancer and CMT tumors were merged and subjected to hierarchical clustering. To estimate correlations between individual CMT transcriptomes and the expression of tumor-initiating cells, we analyzed 154 up- and 339 downregulated genes of CD44+/CD24− tumor-initiating cells, with signature correlation levels calculated as previously described[32].

**Validation of CMT subtypes in human cohort.** Four expression profiles of human breast cancers in public database (GSE17907, GSE20711, GSE25066, and GSE31519) were obtained with clinical outcomes. Patients in the individual cohorts were discriminated into high and low NMF3 with the median of NMF3 scores (i.e., average expression level of genes in NMF3 metagenes). Patient survival of high and low NMF3 was compared using log-rank tests and also for the EMT scores as average expression of genes in EMT/Hallmark MSigDB gene set.

**Tumor microenvironment profiling.** We used the ESTIMATE R package to estimate scores representative of the relative proportion of immune and stromal cells in the admixture of CMT transcriptome data[38]. To infer the relative abundance of tumor-infiltrating immune cells, CIBERSORT was used, with the LM22 set representing 22 immune cell subtypes[40].

**Reporting summary.** Further information on research design is available in the Nature Research Reporting Summary linked to this article.

## Data availability
Raw DNA- and RNA-sequencing data are publicly available in Sequence Read Archive (SRA) with the accession numbers 159481 (DNA-seq) and 159466 (RNA-seq). Gene expression values are available in Gene Expression Omnibus (GEO) with the accession number GSE119810. The lists of germline and somatic mutations are available in Supplementary Data 4 and 5. Single-nucleotide variants and their allele frequencies are available at the Dog Genome SNP Database (DogSD) from the iDOG website (http://bigd.big.ac.cn/idog/).

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

## Acknowledgements

We thank professor Hee-Myung Park (Konkuk University) for reviewing the canine cancer cohort. We also thank Ms. Hein Chun (Yonsei University) for performing quality control of the data. This research was supported by the Bio & Medical Technology Development Program (NRF-2016M3A9B6903439) through the National Research Foundation of Korea (NRF) and No. 2019R1A2C2002315 funded by Korea government, the Ministry of Science and ICT.

## Author contributions

S.K., H.N., J.H.S., and J.H.C. initiated the study and confirmed the design. T.M.K. conducted the main analysis and generated figures and tables. T.M.K. and S.K. finalized main discoveries, drew conclusions and wrote manuscript. I.S.Y. and H.S.K. worked on functional interpretation. S.L. and K.K.K. worked on early genomic data analysis, which is further revised by I.S.Y. and Y.J.H. I.S.Y., D.K., M.K.S., H.N. worked on transcriptomic analysis and subtyping, which is revised by T.M.K. B.J.S. and J.H.S. collected the samples and conducted clinicopathological analysis. All authors read and approved the final draft.

## Competing interests

The authors declare no competing interests.
