## [Peer Review File · Nature Communications]

Reviewers' comments:

Reviewer #1 (Remarks to the Author): Expert in comparative genomics

In their paper Kim et al., describe a genomic and transcriptomic analysis of a large collection of canine mammary tumours revealing a prominent role for the PI3K pathway in the development of these cancers. Overall this is a well performed study. I have several specific comments.

1. I would not use the terminology hBRCA to describe human breast cancers. BRCA1 and BRCA2 are genes involved in a sub-group of human tumours so this terminology is confusing.
2. The authors comment that a subgroup of tumours have a "faster accumulation of mutations" (abstract) but the authors don't know the rate of mutational acquisition just that different groups of tumours had different levels of mutations. It's possible that, for example, the mutations were acquired over time or as a burst.
3. There is a typo on page 4 line 2.
4. The paper doesn't include a germline analysis. i.e. are there any mutations in established human breast and ovarian cancer genes that might explain why these dogs developed cancer? Admittedly the authors have insufficient power to do this analysis genome-wide but they could (for example) overlap pathogenic ClinVar mutations from humans with SNPs/SNVs from the dogs.
5. Can the authors explain or fix the yellow highlighting in Supplementary Table 1?
6. Figure 1 requires some clarification – firstly the gene names need to be better spaced so they are readable. Secondly, the figure legend does not explain what statistical test was used for the analysis in C and D. For the comparisons between benign vs malignant lesions did the authors consider that this could be entirely explained by normal tissue contamination as recently described in human lung cancer?: <https://www.nature.com/articles/s43018-019-0008-8>
7. It is not clear to me how dNdScv was deployed. Did the authors run it as configured for human or did they adjust for the canine genome? The authors would be advised to connect with Inigo Martincorena to discuss this. As configured for human, dNdScv is not optimal for dog.
8. It struck me that there were recurrent mutations in SOX10, HSP90AA1 and NHRNP1. Are the authors really sure that these are not alignment errors? It is striking that these events are so recurrent. Could they be SNPs that got around the filters? Are these all from the same breed of dog?
9. Was the mutational signatures analysis adapted to take into account the mutation opportunity of the dog genome? The methods don't say how the signature analysis was actually performed – was this a de novo signature prediction? On what basis did they assign dog signatures to the human mutational signature catalogue? What were the statistical cut-offs for these assignments? Generally a COSINE >0.85 is required but some thought will be required since the analysis is being performed between species.
10. The authors mention the 30 Sanger mutational signatures but don't cite the Alexandrov paper from which they came.

I enjoyed reading this paper.

Reviewer #2 (Remarks to the Author): Expert in PI3K

I believe the authors of this paper should be congratulated for the uncommon efforts of sharing with the scientific community such a comprehensive atlas of canine genomic and transcriptomic landscape.

Besides describing the main communalities and differences between human breast cancer and canine breast cancer in term of genomic alterations and tumor "transcriptomic" types, the authors' discoveries offer common ground to use spontaneous canine breast cancer as model to study the efficacy of targeted therapy and, concomitantly, increase the therapeutic options of the dogs with cancers.

The only minor comment I have is that in Fig 1B the PIK3CA hotspots are not clearly separated.

Reviewer #3 (Remarks to the Author): Expert in breast cancer genomics

The manuscript by Kim et al has performed whole-exome and transcriptome analyses of 191 spontaneous canine mammary tumors to look at the genomic and transcriptomic characteristics of the disease. Overall, they find a high percentage of PIK3Ca mutations and alterations in the PIK3CA/AKT pathway. Although a large number of cases have been assessed, the observation of PI3K alterations are not novel. However, the authors do identify gene expression subtypes, however these are not validated, so it's difficult to assess how robust these are.

1. It would be useful to have a summary figure comparing the frequencies of identified mutations to that of human breast cancer
2. The authors use gene expression to identify gene expression based CMT subtypes that share molecular and histological features with human breast cancers. The authors should validate these findings in additional published cohorts to validate their presence.
3. The title of the manuscript suggests there is an opportunity for precision therapeutics in dogs. This should be assessed in order to claim this. i.e. test AKT/PI3K inhibitors in cell models of the disease. This would greatly improve the impact of the manuscript. In addition, it would be beneficial to assess activation of these pathways by western blots/ RPPA arrays to prove this. This is something that has not been done in the field.

Reviewers' comments:

Reviewer #1 (Remarks to the Author): Expert in comparative genomics

In their paper Kim et al., describe a genomic and transcriptomic analysis of a large collection of canine mammary tumours revealing a prominent role for the PI3K pathway in the development of these cancers. Overall this is a well performed study. I have several specific comments.

General answer to Reviewer 1):

We deeply appreciate this reviewer's questions and comments to improve our manuscript. Especially, germline variant analysis added great values to our study. Also, we could revisit our somatic mutation list to refine our discovery.

1. I would not use the terminology hBRCA to describe human breast cancers. BRCA1 and BRCA2 are genes involved in a sub-group of human tumours so this terminology is confusing.

Answer to Q1)

We agree with the reviewer that the terminology of hBRCA may not be appropriate. We use the term of 'human breast cancer' instead of hBRCA in the revised manuscript.

2. The authors comment that a subgroup of tumours have a "faster accumulation of mutations" (abstract) but the authors don't know the rate of mutational acquisition just that different groups of tumours had different levels of mutations. It's possible that, for example, the mutations were acquired over time or as a burst.

Answer to Q2)

We appreciate the comment. Since we have no estimates on the mutation rates of CMT genomes, we removed the related descriptions in the Abstract (page 1) and revised the Discussion to describe such possibility (page 27, line 4-7).

3. There is a typo on page 4 line 2.

Answer to Q3)

We appreciate the comment. We corrected the typo.

4. The paper doesn't include a germline analysis. i.e. are there any mutations in established human breast and ovarian cancer genes that might explain why these dogs developed cancer? Admittedly the authors have insufficient power to do this analysis genome-wide but they could (for example) overlap pathogenic ClinVar mutations from humans with SNPs/SNVs from the dogs.

Answer to Q4)

As the reviewer suggested, we added the germline variant analysis in this revision. We thank the reviewer for the suggestion, as this is another important analysis point given our cohort.

As the reviewer mentioned, the size of cohort is insufficient to drive a novel predisposing variants for canine cancer. And also, matching variant to variant between human and canine genome is often unstable due to their different genome annotation levels and structural gaps. But we could at least confirm that *BRCA1* and *BRCA2*, two very well-known germline variants are present in CMT too. We could calculate the prevalence of *BRCA1/2* variants as 5.5%, which is slightly higher than that in human breast cancer (2.9-3.0%), which is varied in the subtypes in human (i.e., higher in TNBC or basal-like). In addition, gene set level analysis showed that genes that are frequently harbor germline predisposing variants in CMT are enriched in the Homology-dependent recombination pathway.

These analyses have been added in the new section 'Germline predisposing variants in CMT' (page 8, line 20 to page 9), including two new tables (Table 1 and Supplementary Table S5) and one figure (Supplementary Figure S1). Corresponding calling procedures are added in Methods (p.31 line 23 to p.32.

line 11). Again, we appreciate the reviewer's valuable comment.

5. Can the authors explain or fix the yellow highlighting in Supplementary Table 1?

Answer to Q5)

We apologize for the typo errors in the manuscript and related files. We fixed the Supplementary Table 1 and carefully reviewed the revised manuscript.

6. Figure 1 requires some clarification – firstly the gene names need to be better spaced so they are readable. Secondly, the figure legend does not explain what statistical test was used for the analysis in C and D. For the comparisons between benign vs malignant lesions did the authors consider that this could be entirely explained by normal tissue contamination as recently described in human lung cancer?: <https://www.nature.com/articles/s43018-019-0008-8>

Answer to Q6)

We appreciate this valuable comment.

First, as recommended, we revised Figure 1 to improve the readability and to address the statistical tests in the legend.

Second, for the contamination issue, we observed that the estimated tumor purity was not significantly different between benign and malignant CMTs ($P = 0.85$, U test) suggesting that the level of normal contamination may not be a major confounding factor in the comparison of benign and malignant CMTs. According the recommended reference, we corrected the TMB for the estimated tumor purity and observed that the tumor purity-adjusted TMB was not significantly difference between benign and malignant CMTs (Supplementary Fig. 2D; $P = 0.44$, U test). We added the new Results in the revised manuscript (page 12, line 5-10).

7. It is not clear to me how dNdScv was deployed. Did the authors run it as configured for human or did they adjust for the canine genome? The authors would be advised to connect with Inigo Martincorena to discuss this. As configured for human, dNdScv is not optimal for dog.

Answer to Q7)

We appreciate the comments. We found that the dNdScv R-package provides an option to build reference on other species ("Using dNdScv in a different species or assembly" by Inigo Martincorena) (<http://htmlpreview.github.io/?http://github.com/im3sanger/dndscv/blob/master/vignettes/buildref.html>) and we followed the instruction to build canFam3.1-specific reference for the analysis. Thus, we believe that our dNdScv analysis is now optimized for the dog.

8. It struck me that there were recurrent mutations in SOX10, HSP90AA1 and NHRNPH1. Are the authors really sure that these are not alignment errors? It is striking that these events are so recurrent. Could they be SNPs that got around the filters? Are these all from the same breed of dog?

Answer to Q8)

We appreciate this important comment. To begin with the results, we concluded those are likely alignment errors, and removed from our final list.

There are many useful utilities or resources to assess alignment errors in human genome, such as VQSR of GATK (Variant Quality Score Recalibration), random-forest-based site filtration from gnomAD, or Hardy-Weinberg Equilibrium-based test. We truly tried our best to rule out erroneous somatic mutation calls in our original manuscript, but there are only limited resources available for canine genome. In this revision, we revisited our filtration criteria and tried to filter out any dubious calls and maintain only high confidence calls.

In the absence of well-established alignment-based filtration, we built a new workflow that assesses the alignment scores at the optimal mapping site (AS) and the suboptimal site (XS). For each SNV site, these scores are calculated for all the passing reads with alternative allele, and averaged over all samples. In genomic positions with unstable alignments, the suboptimal alignment score (XS) is close to the optimal (AS), because their best alignment site is not unique. We filtered out all the initial somatic SNVs, whose XS is greater than or equal to the 80% of their AS. This resulted in filtration of about 1/4 of previous mutation calls, reducing them from 14,650 to 10,855. Also the mutations in the mentioned genes including *SOX10*, *HNRNPH1*, *PTPRH*, *CEACAM28* were removed from our final list. We confirmed that driver mutations on the PI3K-Akt pathways are not affected. The new workflow is added in Methods (p. 32 line 21 to p.33 line 7). According to the new mutation list, we revised the manuscripts and results/Figures with revised mutations. (Fig. 1 and Supplementary Fig 2, 3, 4 and the sample order of Figure 2).

We again thank the reviewer for the suggestion. It is a good opportunity for us to refine the mutation list and to improve the clarity of our study.

9. Was the mutational signatures analysis adapted to take into account the mutation opportunity of the dog genome? The methods don't say how the signature analysis was actually performed – was this a *de novo* signature prediction? On what basis did they assign dog signatures to the human mutational signature catalogue? What were the statistical cut-offs for these assignments? Generally a COSINE >0.85 is required but some thought will be required since the analysis is being performed between species.

Answer to Q9)

We appreciate the comment. We initially tried to identify *de novo* mutation signatures in CMT using two types of signature discovery algorithm (NMF and WTSI). However, the stability measure (Cophenetic score/NMF and signature reproducibility/WTS) point out the minimum of signature numbers (2 for NMF and 1 for WTSI) as the optimum signature number suggesting the 96 trinucleotide matrices of CMT genomes were not well separated into distinct mutation signatures (shown in Supplementary Fig. 4B) or the mutation matrices are homogeneous. We further observed that the identified mutation signatures showed high level of correlation with Signature #1 (e.g., cosine similarity 0.74 ~ 0.86). In addition, the assignment of known mutation signature revealed that the Sig #1 (among 30 cosmic mutation signatures) was prevalent in CMT genomes among the CMT genomes, thus, supporting that the mutations of CMT genomes are homogeneous. We revised the manuscript to address these scenarios in the Result.

10. The authors mention the 30 Sanger mutational signatures but don't cite the Alexandrov paper from which they came.

Answer to Q10)

We thank reviewer for the suggestion We cited the reference in the revised manuscript.

I enjoyed reading this paper.

Reviewer #2 (Remarks to the Author): Expert in PI3K

I believe the authors of this paper should be congratulated for the uncommon efforts of sharing with the scientific community such a comprehensive atlas of canine genomic and transcriptomic landscape. Besides describing the main communalities and differences between human breast cancer and canine breast cancer in term of genomic alterations and tumor "transcriptomic" types, the authors' discoveries offer common ground to use spontaneous canine breast cancer as model to study the efficacy of targeted therapy and, concomitantly, increase the therapeutic options of the dogs with cancers. The only minor comment I have is that in Fig 1B the PIK3CA hotspots are not clearly separated.

General answer to Reviewer 2):

We thank the reviewer for the positive evaluation. As recommended, we revised the Fig. 1B to improve the readability. Also many other points were corrected/revised to improve the overall study. We hope that our

study can be a milestone in applying precision genomics to non-human cancers.

Reviewer #3 (Remarks to the Author): Expert in breast cancer genomics

The manuscript by Kim et al has performed whole-exome and transcriptome analyses of 191 spontaneous canine mammary tumors to look at the genomic and transcriptomic characteristics of the disease. Overall, they find a high percentage of PIK3Ca mutations and alterations in the PIK3CA/AKT pathway. Although a large number of cases have been assessed, the observation of PI3K alterations are not novel. However, the authors do identify gene expression subtypes, however these are not validated, so it's difficult to assess how robust these are.

General answer to Reviewer 3):

We thank the reviewer for the comments. We are also aware of that alterations in PI3K-Akt pathway have been reported in previous studies in non-human cancers. However, no studies has investigated genome-level multi-omics assessments in a large scale to derive the confirmatory landscape of somatic mutations and their prevalence, copy number variations, and pathway level analysis to enable cancer to cancer comparisons. In genomics studies, we believe that quantity can create novelty to provide the community with the macroscopic view of the specific cancer type. We hope that our answers and the revised manuscript could meet the reviewer's criteria for evaluation.

1. It would be useful to have a summary figure comparing the frequencies of identified mutations to that of human breast cancer

Answer to Q1)

We thank the reviewer. As recommended, we include a new table (Supplementary Table S9) to describe the non-silent mutation frequencies of CMT (benign and malignant) and human breast cancers.

2. The authors use gene expression to identify gene expression based CMT subtypes that share molecular and histological features with human breast cancers. The authors should validate these findings in additional published cohorts to validate their presence.

Answer to Q2)

We appreciate this valuable comment. To validate the presence of NMF3 CMT subtypes in human breast cancers, we obtained four public gene expression datasets with patient survival (GSE17907, GSE20711, GSE25066, and GSE31519). We observed breast cancer patients with high NMF3 metagene scores showed unfavorable prognosis with elevated EMT metagene scores. The results are described in revised manuscript (page. 19 line 20 to page 20, line 2) with a new Supplementary Fig 8 in the revised manuscript.

3. The title of the manuscript suggests there is an opportunity for precision therapeutics in dogs. This should be assessed in order to claim this. i.e. test AKT/PI3K inhibitors in cell models of the disease. This would greatly improve the impact of the manuscript. In addition, it would be beneficial to assess activation of these pathways by western blots/ RPPA arrays to prove this. This is something that has not been done in the field.

Answer to Q3)

We appreciate this comment; however, we regret to say that experimental validation of AKT/PI3K inhibitors is beyond the scope of this study. Our manuscript is focused on the identification of the somatic landscape and molecular subtypes in dog cancer, which can be a basis for application of precision medicine strategies to dog cancer. As we have states in the Discussion ("supports the potential use of therapeutics for human breast cancer in dogs with CMTs"), we believe that the remarkable genomic similarities between dog and human may be supporting evidence for the cross-species use of anticancer drugs. We further discuss the discrepancies between animal and human testing reported in clinical trials in the Discussion to address the potential concerns.

REVIEWERS' COMMENTS:

Reviewer #1 (Remarks to the Author):

Overall the authors have done a great job in revising the manuscript. I do, however, have several specific points.

1. The authors could say that some of the BRCA2 variants they found (supp. table 5) were stop codons - I think this is interesting.

2. I am not sure what the authors mean by the "Wellcome Trust Sanger Institute mutation signatures framework"? Do they mean the COSMIC signatures?

3. In Figure 1 there is no scale to explain the degree of gains and losses are the colours binary? I don't think this is the case because there is light blue and dark blue and light red and dark red. etc.

4. In Figure 4 (panel C) the authors use the terminology hBRCA which as discussed previously is confusing. It is corrected in the figure legend but not in the figure.

An interesting paper and useful dataset.

Reviewer #3 (Remarks to the Author):

The authors have mainly addressed the concerns and have highlighted in the discussion regarding point 3 that "AKT/PI3K inhibitors is beyond the scope of this study. Our manuscript is focused on the identification of the somatic landscape and molecular subtypes in dog cancer, which can be a basis for application of precision medicine strategies to dog cancer".

Whilst this is appreciated, as in humans there needs to be thorough testing of therapeutic potential before clinical trials of drugs can begin. I would thus suggest that in the absence of evidence in cell line models that the authors alter the title as they have not directly addressed whether canine tumours with PI3K alterations would benefit from therapeutic targeting of this pathway.

I. Response to reviews

Reviewer #1 (Remarks to the Author):

Overall the authors have done a great job in revising the manuscript. I do, however, have several specific points.

1. The authors could say that some of the BRCA2 variants they found (supp. table 5) were stop codons - I think this is interesting.

- We thank for the reviewer's valuable suggestion. We have added a description about the nonsense mutations of BRCA1/2 at page 9, line 1.

2. I am not sure what the authors mean by the "Wellcome Trust Sanger Institute mutation signatures framework"? Do they mean the COSMIC signatures?

- The Wellcome Trust Sanger Institute (WTSI) mutation signature framework is currently provided as "SigProfiler". Therefore we have added the URL of the SigProfiler in the sentence (Page 11, line 4).

3. In Figure 1 there is no scale to explain the degree of gains and losses are the colours binary? I don't think this is the case because there is light blue and dark blue and light red and dark red. etc.

- We thank for the comment. We think the figure the reviewer mentioned is Fig2A not Fig1. We now added a color bar for the degree of gain and loss of CNVs in the updated Figure 2A.

4. In Figure 4 (panel C) the authors use the terminology hBRCA which as discussed previously is confusing. It is corrected in the figure legend but not in the figure.

- We missed the remaining hBRCA at the lower part of the figure. We now revised the Figure 4. We thank the reviewer for the comment.

An interesting paper and useful dataset.

Reviewer #3 (Remarks to the Author):

The authors have mainly addressed the concerns and have highlighted in the discussion regarding point 3 that "AKT/PI3K inhibitors is beyond the scope of this study. Our manuscript is focused on the identification of the somatic landscape and molecular subtypes in dog cancer, which can be a basis for application of precision medicine strategies to dog cancer".

Whilst this is appreciated, as in humans there needs to be thorough testing of therapeutic potential before clinical trials of drugs can begin. I would thus suggest that in the absence of evidence in cell line models that the authors alter the title as they have not directly addressed whether canine tumours with PI3K alterations would benefit from therapeutic targeting of this pathway.

- We agree with the reviewer's comment that mentioning the opportunity for therapeutics in the title may mislead the readers. Along with the editorial request, we now removed the therapeutics part in the title. The final title is "Cross-species oncogenic signatures of breast cancer in canine mammary tumors".